# Retrospective Cohort Study of the Effectiveness of the Sputnik V and EpiVacCorona Vaccines against the SARS-CoV-2 Delta Variant in Moscow (June–July 2021)

**DOI:** 10.3390/vaccines10070984

**Published:** 2022-06-21

**Authors:** Olga Matveeva, Alexander Ershov

**Affiliations:** 1Sendai Viralytics LLC, 23 Nylander Way, Acton, MA 01720, USA; 2Medusa Project SIA, Krisjana Barona iela 5-2, LV-1050 Rīga, Latvia; anershov@gmail.com

**Keywords:** COVID-19, vaccine effectiveness, SARS-CoV-2, Delta variant, Sputnik V vaccine, EpiVacCorona vaccine, medical informatics

## Abstract

The goal of this study was to evaluate the epidemiological effectiveness of the Sputnik V and EpiVacCorona vaccines against COVID-19. This work is a retrospective cohort study of COVID-19 patients. The cohort created by the Moscow Health Department included more than 300,000 infected people who sought medical care in June and July 2021. Analysis of data revealed a tendency for the increase in the Sputnik V vaccine effectiveness (VE) as the severity of the disease increased. Protection was the lowest for mild disease, and it was more pronounced for severe disease. We also observed a decrease in VE with increasing age. For the youngest group (18–50 years old), the estimated VE in preventing death in June 2021 was 95% (95% CI 64–100), and for the older group (50+ years old), it was 74% (95% CI 67–87). The estimated protection against a severe form of the disease in the 18–50-year-old group was above 81% (CI 95% 72–93), and in the 50+ years-old group, it was above 68% (CI 95% 65–82). According to our analysis, EpiVacCorona proved to be an ineffective vaccine and therefore cannot protect against COVID-19.

## 1. Introduction

A wide range of effective vaccines for the prevention of COVID-19 are now available worldwide. The efficacy and effectiveness of these vaccines have been demonstrated in both clinical and retrospective studies [1,2,3,4]. However, the SARS-CoV-2 viral genomes continue to evolve, giving the virus an increasing number of advantages in evading the immune response. In this regard, the effectiveness of vaccines based on antigens of the ancestral variant needs to be re-evaluated for new viral variants of concern (VOCs) that can cause new epidemic waves.

There are two vaccines, the effectiveness of which is analyzed in this study. Both have been approved for use in Russia. One of them is a viral vector vaccine (Sputnik V), and the other is a peptide-based vaccine (EpiVacCorona).

Sputnik V (Gam-COVID-Vac) is based on a recombinant human replication-defective adenovirus of two types: Ad26 (serotype 26) and Ad5 (serotype 5). Both Ad26 and Ad5 are used as vectors for the expression of the SARS-CoV-2 spike protein (S-protein). Immunization with Sputnik V occurs with two doses of the vaccine, one of which is a primer (rAd26) and the other of which is a booster (rAd5).

There is another vaccine that resembles Sputnik V (primer dose), which was developed by Janssen/Johnson & Johnson. The vaccine Ad26COV2 is based on the rAd26 adenovirus vector. It encodes a full length, stabilized S protein of SARS-CoV-2 and is used in a one-time, single-dose immunization regimen [5].

According to a randomized, double-blind, placebo-controlled, multicenter phase 3 trial that took place in Russia, the protective efficacy of the Sputnik V vaccine is 91.6% [6]. In Hungary, a large retrospective cohort study was carried out to compare the effectiveness of vector-based and mRNA-based vaccines. In this study, almost all the data were gathered before the spread of the Delta VOC. Hungarian researchers estimated the protective effectiveness of Sputnik V in preventing symptomatic infection at 86% and its ability to prevent deaths associated with COVID-19 at 97% [7]. Analysis of data from retrospective case-control studies conducted in Bahrain shows that the ratio of the risks of death in the vaccinated and unvaccinated groups was 1/15. Consequently, the vaccine effectiveness in preventing deaths exceeds 90% [8]. Unfortunately, the publication indicated that during the time when the study was conducted in the country, the dominant variants of the virus changed, so it is not possible to link the estimated vaccine effectiveness to a specific variant of SARS-CoV-2.

Russia has also developed another variant of the Sputnik V vaccine, called Sputnik-Light. This is a single-dose version of Sputnik V, which includes only a primer dose for immunization. The effectiveness of the similar vaccine Ad26COV2. (Janssen/Johnson & Johnson) against COVID-19 infection, based on the results of phase 3 clinical trials, is like Sputnik Light at 67% (95% CI 59–73). The effectiveness of the vaccine against severe COVID-19 is slightly higher at 85% (95% CI 54–97) [5]. However, these trials were conducted before the Delta VOC became dominant. It can be assumed that in a situation with the dominant viral variant Delta or Omicron, the result would be different.

The vaccine representing the second platform in Russia is the peptide vaccine EpiVacCorona. It consists of three peptides of SARS-CoV-2 S-protein conjugated to a carrier protein. The carrier in EpiVacCorona is a two-part chimeric protein, one of which is a viral nucleocapsid protein (N-protein), and the other a bacterial maltose-binding protein. The carrier protein is expressed in *E. coli*, and peptides are also expressed in *E. coli* or chemically synthesized. The three peptides of the vaccine have the following amino acid sequences: CRLFRKSNLKPFERDISTEIYQAGS, CKEIDRLNEVAKNLNESLIDLQE, and CKNLNESLIDLQELGKYEQYIK [9]. It is worth noting that these three peptides do not overlap with the mapped antigenic linear epitopes of the S-protein of SARS-CoV-2 [10,11,12,13,14,15].

EpiVacCorona received a registration certificate on 14 October 2020 [16]. In Russia, the certificate provides emergency use authorization. By the time large-scale immunization with this vaccine began (11 December 2020), even a Phase I clinical trial had not been completed. The representative of the State Research Center “Vector” told the reporter of RIA-Novosti (Russian state media outlet) on 22 January 2021: “Clinical trials of Phases I-II have not yet been completed. There are only intermediate results” [17]. Later, preclinical, and clinical phase I/II trial results were published in Russian journals that are not referenced in PubMed [9,18].

However, these publications and the design of the vaccine itself have come under serious criticism from the scientific community. For example, the lack of important controls in published experiments was noted. In addition, the vaccine has been criticized for the lack of overlap between the three peptides and the experimentally determined linear antigenic epitopes of SARS-CoV-2 S-protein reactive B-cells [19].

After an injection of EpiVacCorona, a vaccinated person can develop antibodies not only to the S-protein peptides of the coronavirus, the protective function of which has not been established, but also to the chimeric protein antigens present in the vaccine to the viral N-protein and bacterial maltose-binding protein. The antiviral immune protective function of the latter has not been demonstrated either.

Independent studies, the results of which were presented on the preprint server in Russia, showed the absence of neutralizing antibodies in the plasma of those vaccinated with EpiVacCorona [20].

Only 3000 participants were enrolled in the EpiVacCorona Phase III clinical trial [21]. It was planned that 25% (750 of 3000) of the participants would receive a placebo. The total number of participants was very small. Therefore, it is difficult to imagine that any statistically significant information can be extracted from the trial data. The trial was registered with ClinicalTrial.gov on 3 March 2021.

At the time of this writing (May 2022), the results of Phase III clinical trials showing the epidemiological effectiveness of the vaccine have not been published. To the best of our knowledge, the first data on the effectiveness of vaccines are presented below.

Delta (B.1.617.2) VOCs were first detected in patient samples from India but quickly spread and became dominant in other countries [22]. This viral variant can circulate efficiently at current vaccination levels in most countries [23,24,25]. In addition, the effectiveness of vaccines against the disease caused by the SARS-CoV-2 Delta variant was shown to be reduced [26,27,28].

In Moscow, the Delta variant replaced all other variants and became dominant in the summer of 2021 [26,29,30]. In June, it averaged more than half of COVID-19 infections, and in July, its share was already more than 90%. However, most of the cases in June corresponded to the second half of the month, when the Delta VOC became dominant. Among the variants of B.1.617.2, the following representatives, AY.4, AY.5, AY.6, AY.10, AY.12, AY.20, AY.23, and AY.24, were encountered in Moscow. However, two variants prevailed: B.1.617.2 and AY.12 [29,31]. The genomes of these variants have several characteristic mutations in the S-protein, which significantly reduce the neutralizing potential of antiviral antibodies directed to this protein [32].

It is of interest to conduct studies that can reveal vaccine effectiveness against different dominant virus variants in Russia. Performing such studies is a new challenge at present since vaccines were developed against one (Wuhan) ancestral variant of SARS-CoV-2, whereas they should protect against other rapidly appearing variants with different antigenic properties. The effectiveness of vaccines registered for use in Russia, especially those with no published results from Phase III clinical trials, such as EpiVacCorona, needs to be evaluated.

One method that can help answer the question of whether a vaccine against the new dominant variant of SARS-CoV-2 is effective is a retrospective cohort data analysis. We performed such a study based on data collected in June and July 2021 in Moscow, Russia. During this period, the viral Delta variant dominated [26,29,30].

## 2. Materials and Methods

### 2.1. Dataset of COVID-19-Infected Individuals

We conducted our analysis using the dataset created by the Moscow Health Department that included people who sought medical care for COVID-19 in Moscow, Russian Federation, in June and July 2021. The dataset was published in the Telegram group “COVID-19 Vaccine News: 2021” and is in the public domain [33].

In this dataset, the number of COVID-19 cases is divided into categories according to patient age and vaccination status. All vaccinated cases were further divided into subcategories according to the type of vaccine (Sputnik V, EpiVacCorona, and CoviVac) and the number of doses (one or two) patients received. In addition, COVID-19 cases are categorized according to the severity of patients’ illnesses or deaths. The severe form of the disease patient group includes those with severe pneumonia, in which more than 75% of the lungs are affected. The moderately severe patient group included those who had less lung damage but still needed oxygen support. Hospitalized patients not requiring oxygen supplementation or outpatients constitute the mild disease group, and PCR+ patients without COVID-19 symptoms constitute the asymptomatic group.

In June, Moscow residents were mainly immunized with Sputnik V. At the end of the month, the number of those immunized with other vaccines was less than 3% of the total number. In July, there were more of those immunized with EpiVacCorona or Covivac. In our work, only those vaccinated with two doses of each vaccine were counted.

A less detailed analysis of Sputnik V data, as well as a comparison of its effectiveness with EpiVacCorona vaccine was carried out for data obtained in July. At the same time, a more detailed analysis of the VE of Sputnik V was carried out using the data obtained in June. We were unable to perform a more detailed analysis of VE values for July data since the number of people vaccinated and diagnosed with COVID-19 for July was available in the database only for the severe COVID-19 case category.

Russian vaccination protocols do not recommend mixing vaccines. Therefore, the entire vaccinated population received only one type of vaccine.

In our work, a few methods were used to count the number of people in a control group. From the total number of city residents in each age group, we subtracted (1) the number of vaccinated individuals or (2) the number of seropositive individuals. The first control group was formed by subtracting vaccinated city residents from the total population of the same age group. The second control group was formed by subtracting seropositive city residents from the total population of the same age group.

To calculate the number of those fully vaccinated by the time the number of cases was estimated, we used data from the Ministry of Health registry created for the registration and issuance of vaccine certificates in Russia [34]. A representative sample of the registry contents was generated by computer polling of vaccine certificate issuance service URL addresses from the space of all possible unique registry record numbers. Data from this registry on those who received Sputnik V or EpiVacCorona vaccines were grouped into the same age categories used to estimate COVID-19 cases.

The 2021 demographics were used to normalize the data and estimate the total number of Moscow residents in different control age groups [35]. The study also used estimates of the total number of vaccinated and COVID-19-infected city residents at various dates in June and July 2021 [36,37].

We calculated the number of seropositive city residents in each age group based on the Moscow Department of Health data in the published preprint [29]. The details are shown below in the following section.

### 2.2. Estimation of the Percentage of Moscow Residents with Antibodies

The monitoring of a representative sample of Moscow residents demonstrated that just under half of them had antibodies to SARS-CoV-2 by the end of June 2021. This number is based on the results of continuous serological studies. For this seroprevalence study, samples were from patients who were admitted to the hospital for routine treatment for a disease not related to COVID-19. The average number of patients tested for SARS-CoV-2 antibodies was 10,000 per week. The presence of IgG antibodies in the venous blood serum was evaluated in all patients using a Mindray Medical International Limited (China) immunochemiluminescent analyzer. The results of sero-monitoring were made public in the preprint publication [29]. This estimation was used by us to form a second control group by subtracting seropositive city residents from the total population.

### 2.3. Method of VE Calculation

The calculations were based on published algorithms [38,39]. The significance values for VE estimations were calculated using the chi-square method [40]. If the outcome in the study population is rare, as with COVID-19 cases among vaccinated or unvaccinated individuals, the odds ratio obtained accurately estimates the risk ratio (RR).

The odds ratio (OR) was calculated as follows:$$\mathrm{OR}=\frac{a\times d}{b\times c}$$
$$95\%\;\mathrm{OR}=\exp(\ln\;\left(\mathrm{OR}\right)\pm 1.96\times \sqrt{\frac{1}{a}+\frac{1}{b}+\frac{1}{c}+\frac{1}{d}})$$
considering:number of COVID-19 cases among vaccinated people in Moscownumber of vaccinated in Moscownumber of COVID-19 cases among unvaccinated people in Moscownumber of Moscow residents in the control group. To calculate the number of individuals in the control group, two variants of numerical estimation were used for each age group: (1) the number of unvaccinated Moscow residents and (2) the number of seronegative city residents.

Vaccine effectiveness (VE) % was calculated as follows:VE = 100 × (1 − OR)

The data for the calculations were obtained from the following sources:

The total number of Moscow residents in each age group was calculated based on the city’s 2021 demographics [35].

a and c—[33] and Appendix A.

b.register of vaccinations in Moscow, with data aggregated for the same age groups [34] and [36].d.A numerical estimate of the number of individuals in each age group of the control group was made by subtracting the number of vaccinated or seropositive citizens from the number of Moscow residents in each age group.

### 2.4. Confidence Intervals for Vaccine Effectiveness Estimates

In this paper, we estimate vaccine effectiveness using two types of assumptions about the control group. Because of this, we can only estimate a vaccine’s effectiveness value as a value within a confidence interval whose width is determined by our assumptions. Thus, the lower end of the interval for estimating effectiveness can be used as the minimum among all estimates derived from assumptions about the size of the control group, and the upper end can be used as the maximum. The confidence intervals for the effectiveness of the Sputnik V vaccine in preventing severe disease, calculated using this algorithm, are shown in the last column of Table 1.

## 3. Results

### 3.1. The Effectiveness of Sputnik V and EpiVacCorona against Severe COVID-19 and Death

The results of the VE analysis of the Sputnik V and EpiVacCorona vaccines against severe COVID-19 are shown in Figure 1a. A comparison of the results in the upper and lower bar charts of Figure 1a suggests that, regardless of the definition of the control group, the estimated VE of Sputnik V to prevent severe COVID-19 is high, and its value is statistically significant. The bar graphs also show the presence of an age-related decrease in vaccine efficiency. The older the person is, the lower the estimated VE is.

Estimates of the number of deaths and severe COVID-19 among vaccinated or seronegative people in Moscow are plotted on the bar graphs shown in Figure 1b. We clearly see that the risk of either outcome is drastically reduced in individuals vaccinated with Sputnik V. Among those who received the vaccine but were older than 51, the number of deaths and serious illnesses was significantly higher compared to those who were younger than 50.

We also created a graphical illustration showing the Sputnik V VE in preventing COVID-19-related deaths (Figure 2, left panel). A comparison of the upper and lower charts in the left panel of Figure 2 shows that the estimate of VE in preventing COVID-19 deaths is almost independent of how the control group is defined.

More detailed information on the age-related VE of Sputnik V is presented in Table 1. Because any control group definition is not completely satisfactory in all necessary aspects of VE estimation, it may be appropriate to use a combination of calculated 95% confidence intervals to determine the upper and lower limits of VE. The combination obtained from the two control groups is presented in the last column under the title “CI 95% of both control groups” in Table 1. Based on this 95% CI combination, we can say that the vaccine protects against severe forms of the disease with an effectiveness of over 81% in the group under 50 years of age and over 32% in the group over 70 years of age.

In Figure 1a, instead of positive estimates of EpiVacCorona vaccine effectiveness, negative estimates are shown. Despite the wide confidence intervals of these negative values, we can confidently state that the EpiVacCorona vaccine cannot protect against severe COVID-19.

### 3.2. Comparative Effectiveness of the Sputnik V Vaccine in Preventing COVID-19 of Varying Severity in Different Age Groups

Table 2 demonstrates the calculated odds ratios of COVID-19 outcomes among vaccinated or seronegative individuals. Although all calculations were based on data collected in one month (June or July), we introduced a time parameter into our estimates to compare our results with those of other studies that use time normalization.

Figure 1 and Figure 2 show a tendency for the calculated VE values to decrease as age increases. This trend was independent of which control group was used for the analysis. According to this estimation, breakthrough infections occur more frequently in the older age group.

In addition, Figure 2 also revealed the following trend: the more severe the disease, the better the vaccine protection observed. We noticed a tendency for vaccine effectiveness to increase as the severity of the disease increases, so that the vaccine protects particularly well against the most severe form of the disease or even death.

## 4. Discussion

### 4.1. Limitations of Our Study

Our retrospective cohort study has several limitations. We must deal with a dataset that lacks important information about those who received COVID-19. We do not know the demographics, except for the age of COVID-19 patients. We do not know the timing of vaccination, and we cannot connect each vaccinated and unvaccinated person so that their characteristics match each other. Commercial test systems that have been used to determine the presence of antibodies in the Moscow population have not been able to distinguish antibodies acquired through natural infection from those acquired through vaccination against SARS-CoV-2. Therefore, evidence of prior infection was not available for the studied population. However, estimates of the proportion of the Moscow population with antibodies capable of recognizing viral proteins were available (at the end of June). Nevertheless, it was unclear whether these antibodies recognized viral N-protein, which arise because of natural infection, or S-protein, which can also arise as a result of vaccination.

### 4.2. EpiVacCorona Is an Inefficient Vaccine

To date, the developers of EpiVacCorona have not published the results of their phase III study. The results of this analysis, along with an earlier analysis [41], indicate a lack of protective VE. Our study showed the negative efficacy of EpiVacCorona. What could this be due to? We assume that the active or even aggressive advertising campaign in Moscow for this particular vaccine as the most sparing for health, i.e., the vaccine with the least number of side effects, led to a greater number of people with morbid conditions choosing this particular vaccine. Thus, it is highly likely that there were many more chronically ill people in the vaccination group who were prone to more severe COVID-19. As a result of this bias, the vaccinated group has more incidents of severe disease and deaths compared to unvaccinated groups.

Of course, one can also try to explain the negative effectiveness of EpiVacCorona by biological factors. For example, immune imprinting may be to blame. The vaccine contains a large amount of viral N-protein. Thus, when those who have been immunized with EpiVacCorona encounter a real virus, their memory B-cell clones produce mainly IgG antibodies targeting the viral N-protein. These antibodies cannot bind the virus, since only antibodies targeting the viral S-protein are able to do that and induce immune protection. Therefore, the immunized person’s body produces the wrong ineffective antibodies after SARS-CoV-2 invasion.

The developers of EpiVacCorona tried, but failed, to equip the vaccine with S-protein fragments (in the form of three peptides), which can trigger an antibody response to the virus. This failure is explained by a lack of overlap between the three protein fragments they chose as vaccine peptides and the experimentally mapped linear antigenic epitopes of the S-protein of SARS-CoV-2 [10,11,12,13,14,15].

### 4.3. Perhaps There Are More People with Comorbid Chronic Conditions in the Vaccinated Elderly Groups

It may be assumed that the vaccinated elderly groups had many more people with comorbid chronic conditions than the other groups, simply because people with chronic conditions were more motivated to get vaccinated and received the vaccine more often. As a result, despite vaccination, groups of older people with more comorbid conditions had more infections than they would have had if people with chronic conditions were not there. This effect can significantly decrease the VE in certain age groups of people or even make it negative. Perhaps for this reason, and because there are more people with comorbid chronic diseases in the vaccinated elderly groups than in other groups, morbidity after EpiVacCorona is somewhat more common among vaccinated individuals and results in negative vaccine efficiency values. It is possible that this may also explain the low or even negative efficiency values of Sputnik V for preventing non severe forms of COVID-19 disease in the older age groups.

### 4.4. Biological Factors May Contribute to the Decrease in VE in Older Age Groups

In addition to social and behavioral factors, features of the immune system of the elderly may contribute to the age-related decline in VE. Below is a detailed description of the specific functioning of the immune system in the elderly.

#### 4.4.1. Protection against SARS-CoV-2 Reinfection Is Lower in the Older Age Group

Interestingly, the ability of the human immune system to protect against COVID-19 reinfection also appears to depend on age. For example, a study of 4 million RT-PCR-positive cases in the first and second wave of infections in Denmark showed that protection against reinfection was stronger in the younger generation: 81% in the under-65 group versus 47% in the over-65 group [42].

#### 4.4.2. The Rate of Decrease in the Level of Antibodies Is Higher in the Older Age Group

Our retrospective cohort analysis revealed a consistent decline in the estimated vaccine effectiveness with age. Although we attribute this decline to the assumption that there are more people with comorbidities among the elderly vaccinated compared to the unvaccinated of the same age, we do not rule out that in the elderly, vaccine-induced protective immunity can weaken more quickly because their antibody levels might decrease faster, as shown in a number of publications for different vaccines [43,44,45,46]. After vaccination with BNT162b2, the level of antiviral IgG antibodies decreased faster in older vaccine recipients (≥50). Interestingly, no such effect was observed after mRNA-1273 vaccination for six months [47].

### 4.5. Comparison with Retrospective Studies of Vaccine Effectiveness Estimates in Different Countries

In general, our VE estimate of the Sputnik V vaccine in preventing COVID-19-related deaths or severe disease is comparable to the VE estimates of vaccines from other developers. However, the VE that we estimated decreased as the age of the individual increased. The reasons for this trend may be related to deficiencies in the cohort study design that does not consider large differences in the characteristics of people in the observation groups. In addition, differences in the immunological and behavioral characteristics of people in different age cohorts may contribute to this effect. Does the same tendency appear in the studies of other authors? As seen in Table 3, the results are variable but most frequently are consistent with our own.

### 4.6. Perspectives. How Long Does Vaccine Protection Last?

How long can Sputnik V protect against severe COVID-19 and death? When should the boosting dose be given? There are no open databases that can be analyzed to answer these questions. However, studies have been performed with other vaccines.

An interesting retrospective cohort study of the duration of vaccine protection against infection was conducted in Sweden [49]. It was found that the effectiveness of the vaccine in protecting against COVID-19 infection decreased faster among men and the elderly. The observations of this study, in terms of decreasing vaccine efficacy with age, are consistent with our findings.

In the Swedish study, however, special emphasis was placed on examining the duration of the protective effect of the vaccine. It was shown that, on average, over a period slightly longer than six months, vaccine efficiency in preventing symptomatic COVID-19 for the BNT162b2 vaccine dropped from 92% to zero. Additionally, over an even shorter period (four months), the efficiency of the ChAdOx1 nCoV-19 vaccine dropped from 66% to zero. Slightly more stable was the efficiency of the mRNA-1273 vaccine (Moderna). It dropped from 96% to 59% in half a year [49]. However, the average efficiency of all vaccines to prevent symptomatic COVID-19 for those under age 50 was still positive, although not high (~34%) even after half a year. In contrast, for those over 50 years of age, vaccine efficiency in just over half a year waned to zero [49].

Interestingly, in a retrospective cohort study in the United States, the results showed a slightly slower decline in the effectiveness of the BNT162b2 vaccine compared to the decline found by scientists in Sweden. The protective effectiveness (against infection with the Delta variant of SARS-CoV-2) in the United States fell from 93% to 53% in five months [53] and not to zero as in Sweden [49]. However, the VE against Delta-related hospitalizations was overall high at 93% and did not decline for at least six months. VE was age dependent and was higher for younger adults [53].

Similar results were obtained in the UK in a case-control study. VE against symptomatic disease caused by the Delta viral variant dropped to 70% for BNT162b24 and to 47% for ChAdOx1 nCoV-19 over 20 weeks but did not fall as much for hospitalizations. The decline in the efficiency of these vaccines was more noticeable in people over 65. The authors conclude that the VE declines much faster in older people than in younger people [50].

British researchers performed a retrospective cohort design study [52] and presented an analysis of a large database obtained mainly from the United States. In this database, the researchers identified more than 10,000 vaccine breakthrough COVID-19 cases, which were matched with those of unvaccinated controls. The work showed that the vaccine protected those under 60 years of age from death and intensive care unit hospitalization significantly better than those who were older. In fact, the authors of the study did not detect any positive effects of vaccination for those over 60 years of age [52]. Unfortunately, the type of vaccine was not specified in the study.

Finally, an Israeli study demonstrated that immunity against SARS-CoV-2 infection waned in all age groups a few months after vaccination. However, after the same period following the second dose of the vaccine, severe COVID-19 was more common among those over 60 years of age [54].

In summary, it is unlikely that the Sputnik V vaccine protects longer than the vaccines mentioned above, so a boosting dose can be useful six months after the first vaccination. This recommendation is most applicable to the elderly.

## 5. Concluding Remarks

Our retrospective study shows that those vaccinated with EpiVacCorona have no advantage over unvaccinated or seronegative individuals in their chances of contracting COVID-19. The vaccine was introduced into civilian circulation without sufficient testing, has not proven itself, and should therefore be withdrawn from production and distribution. At the same time, all those immunized with this poorly tested vaccine preparation should be given the opportunity to be vaccinated with a modern, effective vaccine.

We find that the estimated VE of the Sputnik V vaccine to prevent deaths and severe forms of disease caused by the Delta variant of SARS-CoV-2 is comparable to the estimated VE of this vaccine against COVID-19 caused by other viral variants that have previously appeared in circulation. Our observations are consistent with other studies that have evaluated the VE of Sputnik V in clinical trials [6] or case-control studies [41]. However, due to several factors, and primarily due to the limitations of the database with which we worked, it was impossible to directly compare the estimated VE values in our work with VE estimates from published papers. Our database does not allow us to normalize the analyzed groups by the number of people with chronic diseases, as well as by other important demographic characteristics. If there are more people with comorbidities among the vaccinated elderly than among the unvaccinated, then the VE calculations will underestimate the effectiveness of the vaccine.

We found that the estimated VE of the Sputnik V vaccine decreases with age and reaches a minimum in the age cohort over age 70. We do not know the exact reasons for this decrease in effectiveness, and several hypotheses have been discussed above. Among them are those based on (a) limitations in the design of our database, (b) biological factors, and (c) social factors.

Biological hypotheses relate to the immune characteristics of the elderly. These characteristics may manifest as lower antibody levels that appear after vaccination in the elderly and/or a more rapid decline in antibody levels. The protective effect of the vaccine may thus be shorter. These hypotheses are partially supported by the literature [43,44,45,57].

A hypothesis based on social factors links the decrease in vaccine effectiveness to behaviors leading to additional risks of COVID-19 infection predominantly in the elderly group. Members of this group might feel more protected and spend more time in public places where the risk of infection is higher, such as on public transportation. The opposite hypothesis can also be considered. It may be that young people, because of their busier social lives compared to older people, are more likely to be infected with COVID-19 in an asymptomatic form, and after vaccination they develop hybrid immunity, which protects better than just vaccine immunity in older people. Hybrid immunity forms in those who have had the disease and have been vaccinated [51].

In any case, we and others have found a pattern of decreasing estimated VE with increasing age. Similar observations for other vaccines have been made in various countries, including Sweden [49], the UK [50,51,58], the United States [52,53], Israel [54], and Qatar [56]. Regardless of how the nature of the detected effect is explained, it might indicate that elderly people get sick more often than younger people, which means that they need to protect themselves to a greater extent, e.g., by booster doses of vaccines and, in addition, by masks, social distance, and all other means.

## 6. Conclusions

In summary, we can state the following:The EpiVacCorona vaccine does not protect against COVID-19.The failure of EpiVacCorona to protect those who have been vaccinated demonstrates that no vaccine should be introduced to the public until strong evidence shows that it is safe and effective.The Sputnik V vaccine appears to confer high (significant) protection against the Delta variant.The more severe the COVID-19 disease, the better the Sputnik V vaccine protects against it.The estimated VE of Sputnik V was lower in the elderly than in the young.

## Figures and Tables

**Figure 1 vaccines-10-00984-f001:**
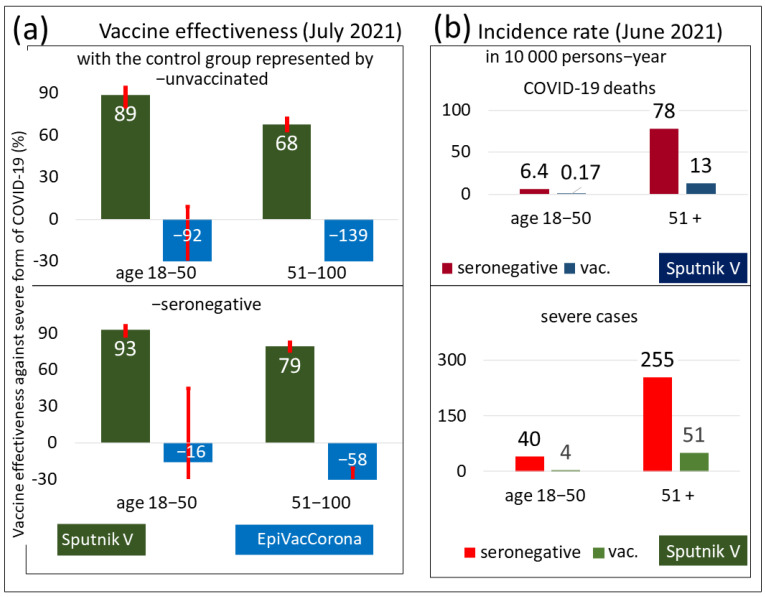
Characterization of immune protection by COVID-19 vaccines. Data obtained in Moscow for the summer of 2021. (**a**) The estimated effectiveness values of the Sputnik V and EpiVacCorona vaccines in preventing severe forms of COVID-19. The analysis is based on data from July 2021. The upper chart shows the results of the analysis performed with a control group of unvaccinated Moscow residents, and the lower chart corresponds to the data analysis performed with a control group of seronegative individuals. The VE estimates for the Sputnik V vaccine were positive and highly significant (*p* < 0.001) by the chi-square test for both age groups. The VE values for EpiVacCorona vaccines were negative and nonsignificant (*p* > 0.05) for the 18–50 age group and negative and significant for the 50+ age group (*p* < 0.001). (**b**) The panel shows the number of deaths and severe cases of COVID-19 among fully vaccinated or seronegative Moscow residents, normalized per 10,000 person-years.

**Figure 2 vaccines-10-00984-f002:**
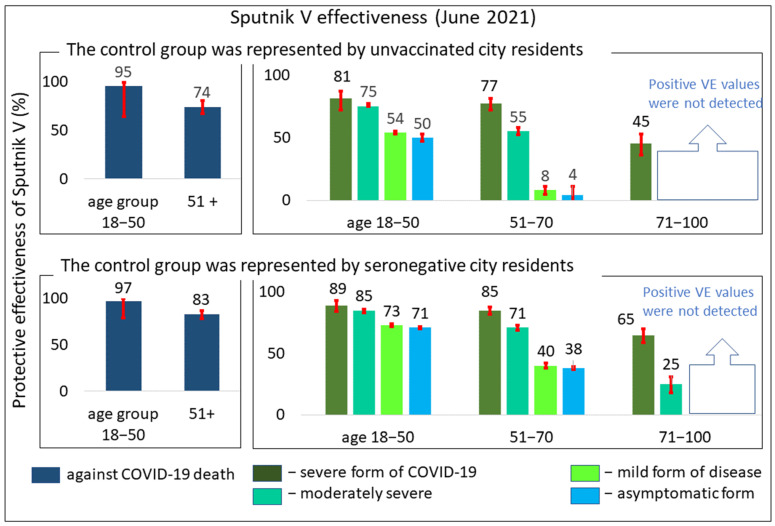
Protective effectiveness of the Sputnik V vaccine against COVID-19 disease of varying severity or death (%). The upper bar charts represent calculations with control group 1, and the lower bar charts represent calculations with control group 2. All positive VE estimates in all charts are highly significant according to the chi-square test, *p* < 0.001. Data obtained in Moscow in June 2021.

**Table 1 vaccines-10-00984-t001:** Efficiency of the Sputnik V vaccine to prevent severe forms of COVID-19 with confidence intervals. Data obtained in Moscow in July 2021.

Age, Years	Control Group—Unvaccinated			Control Group—Seronegative			CI 95% of Both Control Groups	
	VE%	CI 95%		VE%	CI 95%			
		Lower	Upper		Lower	Upper	Lower	Upper
18–50	89	81	94	93	88	96	81	96
51–70	76	70	81	84	80	87	70	87
70+	42	32	51	63	56	69	32	69

VE, vaccine effectiveness; CI, confidence interval. The EpiVacCorona vaccine is not effective.

**Table 2 vaccines-10-00984-t002:** Estimation of deaths and severe COVID-19 cases. Data obtained in Moscow in June 2021.

Age	Normalized Data (10,000 Persons per Year)				Odds Ratio		95% CI			
	Deaths		Severe Form of COVID-19				Deaths		Severe Form of COVID-19	
	Unvac.	Vac.	Unvac.	Vac.	Deaths	Severe Form	Lower	Upper	Lower	Upper
18–50	6.4	0.17	40	4	0.03	0.1	0	0.21	0.07	0.15
51+	78	13	255	51	0.17	0.2	0.13	0.22	0.18	0.23

**Table 3 vaccines-10-00984-t003:** Studies demonstrating the negative correlation between VE and age.

Country of Data Origin	Type of Research	Vaccines	Outcomes of COVID-19	Age Dependence of VE	Virus Variant	Type of Publication	Ref.
Scotland	retrospective case control	BNT162b2, ChAdOx1 nCoV-19	death	not detected	mainly Delta	article	[48]
Sweden	retrospective cohort	BNT162b2, ChAdOx1 nCoV-19, mRNA-1273	death, hospitalizations, symptomatic and asymptomatic	detected	not specified	article	[49]
Hungary	retrospective cohort	BNT162b2, mRNA-1273, Sputnik V, Sinofarm, ChAdOx1 nCoV-19	infection	not detected	not specified	article	[7]
UK	case-control study	BNT162b2, ChAdOx1 nCoV-19	mild and severe	detected	Delta	article	[50]
UK	case-control study	BNT162b2, ChAdOx1 nCoV-20	PCR-positive	detected	Delta	article	[51]
USA	retrospective cohort, controlled by those who got flu vaccine	is not specified	death, ICU hospitalizations, and many other outcomes	detected	not specified	article	[52]
USA	retrospective cohort	BNT162b2	PCR-positive and hospitalization	detected, but minor	Delta	article	[53]
Israel	case-negative control	BNT162b2	PCR-positive or severe form of COVID-19	not detected for PCR+ and detected for severe COVID-19	not specified	article	[54]
Israel	case-negative control	BNT162b2, mRNA-1273	infection	detected but positive	Delta	article	[55]
Qatar	case-negative control	BNT162b2, mRNA-1273	symptomatic or asymptomatic infection	detected	Delta	article	[56]
Russia	retrospective cohort	Sputnik Light	symptomatic infection	detected	Delta	preprint	[29]

## Data Availability

The data used for the calculations made in this work are available in Appendix A. In addition, all input data, calculations and the code are available in GitHub [59]. The content of the manuscript has previously appeared online in ResearchSquare preprint [60].

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
