# Peer review of "Retrospective Cohort Study of the Effectiveness of the Sputnik V and EpiVacCorona Vaccines against the SARS-CoV-2 Delta Variant in Moscow (June–July 2021)"

_vaccines, 2022, doi:10.3390/vaccines10070984_

Round 1

Reviewer 1 Report

The report deals with real-life effectiveness of two COVID-19 vaccines create in Russia that created controversy mostly because of the way they were introduced into mass immunization campaigns. The results of the study that to our knowledge is the first comparison of the two vaccines, clearly demonstrated that one of them, EpiVac Corona is totally ineffective, while the other (Gam-COVID-Vac) is comparable to similar products developed by other companies. Therefore the paper clearly deserves publication, after authors address some issues. 

The results demonstrate that EpiVac Corona did not protect against COVID-19.  It would be important to at least speculate about the reasons why it appears to have negative VE. Is this a car of vaccine indices enhanced disease, or is related to the social behavior of "vaccinated" individuals?

Both Gam-COVID-Vac and EpiVac Corona were hastily registered before all clinical trials were completed. In case of Gam-COVID-Vac there was only phase 1 study with a limited number of mostly young volunteers, while for EpiVac Corona there was no clinical data available at the time of "registration". This gamble paid off in the case of Gam-COVID-Vac  but badly failed for EpiVac Corona. In the opinion of this reviewer the most important conclusion of the paper is that no vaccine must be introduced until solid data proves its safety and efficacy. While this is widely accepted by most regulatory agencies around the world, it apparently is not the case in Russia and possibly some other countries. It would be important to make this point more clearly. 

Authors may want to tone down the language that in some places could be viewed as subjective. For instance, on line 78 they use words "obscure Russian Journal". It may be sufficient to say "Russian-language journal not reverenced in Pubmed". Similarly, on line 397 the word "pseudovaccine" is a bit emotional. Would "untested preparation" be a better choice?

Throughout the paper authors use the word "histogram", while in most cases they refer to bar graphs, which is not the same. The word histogram should refer to bar graphs that show a distribution of a certain variable. Most figures in this paper show comparison between two or more discrete categories. 

Finally, the meaning of the word "successful" on line 29 is unclear. Did you mean "successive"?

Author Response

We are grateful to both reviewers for a very thorough reading of the manuscript and very constructive comments and suggestions. We revised our manuscript as suggested, and all changes are marked. The "Conclusion" section has been revised. 

Reviewer 1

  1. The results demonstrate that EpiVac Corona did not protect against COVID-19. It would be important to at least speculate about the reasons why it appears to have negative VE. Is this a car of vaccine indices enhanced disease, or is related to the social behavior of "vaccinated" individuals?

Answer: we addressed this important question and introduced new text into the revised version of Discussion section of our manuscript (lines 307-314). The following is additional text that was added at the suggestion of the reviewer:

“To date, the developers of EpiVacCorona have not published the results of their phase III study. The results of this analysis, along with an earlier analysis [41], indicate a lack of protective VE. Our study showed negative efficacy of EpiVacCorona. What could this be due to? We assume that the active or even aggressive advertising campaign in Moscow of this particular vaccine as the most sparing for health, the vaccine with the least number of side effects, led to a greater number of people with morbid conditions choosing this particular vaccine. Thus, it is highly likely that there were many more chronically ill people in the vaccination group who were prone to more severe COVID-19. As a result of this bias, the vaccinated group has more incidents of severe disease and deaths compared to unvaccinated groups.”

2. In the opinion of this reviewer the most important conclusion of the paper is that no vaccine must be introduced until solid data proves its safety and efficacy. While this is widely accepted by most regulatory agencies around the world, it apparently is not the case in Russia and possibly some other countries. It would be important to make this point more clearly.

Answer: We added this important conclusion in our list of conclusions (lines 455-457). The following is a list of conclusions with item #2 added.

Conclusions

In summary, we can state the following:

  1. The EpiVacCorona vaccine does not protect against COVID-19.
  2. The failure of EpiVacCorona to protect those who have been vaccinated demonstrates that no vaccine should be introduced to the public until strong evidence shows that it is safe and effective.
  3. The Sputnik V vaccine appears to confer high (significant) protection against the Delta variant.
  4. The more severe the COVID-19 disease, the better the Sputnik V vaccine protects against it.
  5. The estimated VE of Sputnik V was lower in the elderly than in the young.

3. Authors may want to tone down the language that in some places could be viewed as subjective. For instance, on line 78 they use words "obscure Russian Journal". It may be sufficient to say "Russian-language journal not reverenced in Pubmed". Similarly, on line 397 the word "pseudovaccine" is a bit emotional. Would "untested preparation" be a better choice?

Answer: We followed the advice of the review and substitute the emotional words with more appropriate for scientific texts (lines 78 and 412).

4) Throughout the paper authors use the word "histogram", while in most cases they refer to bar graphs, which is not the same. The word histogram should refer to bar graphs that show a distribution of a certain variable. Most figures in this paper show comparison between two or more discrete categories.

Answer: We substituted our incorrectly used term "histogram" with correct ones alternating the words: bar graphs, panels, or bar charts (lines 226, 228, 231,255,256,257). Sometimes we omitted the incorrect word without substitution because the text was sufficiently clear without extra specifications (lines 283, 286).

5) Finally, the meaning of the word "successful" on line 29 is unclear. Did you mean "successive"?

Answer: Were removed the word that was used by mistake (line 29).

Reviewer 2 Report

Matveeva and Ershov determined the effectiveness of the Sputnik V and EpiVacCorona vaccines against the SARS-CoV-2 Delta variant through a retrospective cohort study in Moscow in the summer of 2021.

Although this study is a retrospective cohort data analysis, it is important because it provided us an important piece of information on the effectiveness of the vaccine against an emerging viral variant of concern. In addition, providing detailed information on EpiVacCorona in the introduction is commendable.

The manuscript would be improved if the authors addressed the following points;

1)    The reviewer understands data on the demographics, except for the age of COVID-19 patients or the timing of vaccination, are not available. However, it should be clarified if data on evidence of previous infection or serologic (or virologic) evidence of SARS-CoV-2 infection prior to vaccination were available for the vaccinated population. Please clarify if such data was available. If not, please add this limitation in the discussion section.

2)    Is there any mix and match vaccination in the study population? Or did all the vaccinated population receive one kind of vaccine only? If not, were cases with mix and match vaccination excluded from the study?

Author Response

We are grateful to both reviewers for a thorough reading of the manuscript and very constructive comments and suggestions. We revised our manuscript as suggested, and all changes are marked. 

Reviewer 2.

The manuscript would be improved if the authors addressed the following points:

  • The reviewer understands data on the demographics, except for the age of COVID-19 patients or the timing of vaccination, are not available. However, it should be clarified if data on evidence of previous infectionor serologic (or virologic) evidence of SARS-CoV-2 infection prior to vaccination were available for the vaccinated population. Please clarify if such data was available. If not, please add this limitation in the discussion section.

Answer: We have added text that meets the reviewer's concerns (lines 296-303). This text is below: “Commercial test systems that have been used to determine the presence of antibodies in Moscow population have not been able to distinguish antibodies acquired through natural infection from those acquired through vaccination against SARS-CoV-2. Therefore, evidence of prior infection was not available for the studied population. However, estimates of the proportion of the Moscow population with antibodies capable of recognizing viral proteins were available (at the end of June). Nevertheless, it was unclear whether these antibodies recognized viral N-protein, which arise because of natural infection, or S-protein, which can arise also as a result of vaccination.

2 ) Is there any mix and match vaccination in the study population? Or did all the vaccinated population receive one kind of vaccine only? If not, were cases with mix and match vaccination excluded from the study?

Answer: The following is additional phrase that was added at the suggestion of the reviewer (lines 152-153): “Russian vaccination protocols do not recommend mixing vaccines. Therefore, the entire vaccinated population received only one type of vaccine.”
